# Early Renal Remission Is Associated with Increased Likelihood of Subsequent Remission in Lupus Nephritis: Single-Centre Observational Study in Australia

**DOI:** 10.3390/ijms26199634

**Published:** 2025-10-02

**Authors:** Shiori Nakagawa, Emily K. Yeung, Alberta Hoi, Eric F. Morand, Joanna R. Kent, Rangi Kandane-Rathnayake

**Affiliations:** 1Rheumatology Research Group, Centre for Inflammatory Diseases, School of Clinical Sciences, Monash University, Clayton, VIC 3168, Australia; shiori.nakagawa@monash.edu (S.N.); eric.morand@monash.edu (E.F.M.); 2The George Institute for Global Health, University of New South Wales, Sydney, NSW 2052, Australia; 3Department of Nephrology and Transplantation, Guy’s and St Thomas’ NHS Foundation Trust, London SE1 7EH, UK; 4Department of Nephrology, Monash Health, Clayton, VIC 3168, Australia; joanna.kent@monash.edu; 5Department of Rheumatology, Monash Health, Clayton, VIC 3168, Australia; alberta.hoi@monash.edu

**Keywords:** systemic lupus erythematosus, lupus nephritis, proteinuria, remission

## Abstract

Reliable clinical markers for predicting sustained renal remission remain poorly understood in patients with lupus nephritis (LN). We investigated whether achieving complete renal remission (CRR) within 6 months of induction therapy, compared to within 12 months, was associated with a higher likelihood of attaining CRR at 24 months. We conducted a retrospective observational study of biopsy-proven patients with class III or IV ± V LN treated at a lupus clinic in Australia. CRR was defined as a urine protein: creatinine ratio (UPCR) of <0.05 g/mmol with no worsening of eGFR > 10% from baseline. CRR responders at 6, 12, and 24 months were determined. Associations between 6- and 12-month CRR status and 24-month CRR were examined using logistic regression. In total, 60 patients were included; 49 (82%) were female, with a median age of 27 years (IQR: 19, 39) at LN diagnosis. CRR was attained at 6, 12, and 24 months by 23 (40%), 26 (47%), and 24 (44%) of patients, respectively. Both 6- and 12-month CRR attainment were significantly associated with an increased likelihood of CRR achievement at 24 months (adjusted odds ratios 11.23 (95% CI 2.53, 49.88) and 11.39 (95% CI 2.41, 53.80), respectively). Achieving CRR at 6 and 12 months was a strong independent predictor for attaining subsequent renal remission.

## 1. Introduction

Lupus nephritis (LN) affects approximately 30–50% of patients with Systemic Lupus Erythematosus (SLE) and is associated with an increased risk of kidney failure [1]. Several studies have demonstrated that a reduction in proteinuria and stabilisation of estimated glomerular filtration rate (eGFR) with treatment are associated with better kidney outcomes [2,3,4]. Two retrospective observational studies reported that modified primary efficacy renal response (mPERR; defined as eGFR ≤ 20% below baseline value or ≥60 mL/min/1.73 m^2^ and urine protein: creatinine ratio (UPCR) ≤ 0.7 g/day) achievement at 24 months after biopsy-proven class III or IV ± V LN was associated with lower mortality and end-stage kidney disease (ESKD) [5,6].

Novel therapeutic options have been developed to improve renal prognosis in LN patients, particularly over the past 5 years [7,8]. The recently updated American College of Rheumatology (ACR) guidelines recommend triple immunosuppressive therapy upfront for patients with active class III or IV LN, intensified from the dual therapy recommended in the previous guideline iteration, highlighting the importance of achieving disease control as early as possible [9]. Of equal importance, however, is achieving sustained renal remission. An Italian study of LN patients with a median follow up of 15.7 years demonstrated that 3 years of sustained clinical remission (eGFR 60 mL/min/1.73 m^2^ and proteinuria <0.5 g/24 h) prevented the development of chronic kidney disease (CKD, defined as sustained eGFR < 60 mL/min/1.73 m^2^ for at least 3 months), with a 20-year CKD-free survival of 96% in those who achieved a minimum of 3 years of sustained clinical remission compared to 68% in those who did not [10]. However, reliable clinical markers for predicting sustained renal remission remain poorly understood in patients with LN. This study aims to determine whether early renal remission (ERR) is associated with subsequent sustained renal remission.

## 2. Results

### 2.1. Patient Characteristics at Baseline

Of the 369 patients with SLE treated at the Monash Health Lupus Clinic, a tertiary-level hospital servicing the South-East of Melbourne, Australia, and enrolled in the Australian Lupus Registry and Biobank (ALRB) between 2007 and 2020, 60 (16%) patients had a new diagnosis of kidney biopsy-confirmed class III or IV ± V LN and comprised the study cohort. Baseline characteristics of the study cohort are described in Table 1. Overall, 49 (82%) of patients were female, with a median age of 27 [interquartile range (IQR): 19, 39] years at diagnosis. Overall, 31 patients (52%) were Asian, and 21 (35%) were Caucasian. The median SLE disease duration prior to LN diagnosis was 4.5 [IQR: 0, 10.5] years. In total, 12 patients (20%) had class III, 26 (43%) patients had class IV, and 22 (37%) had class III or IV + V LN. Median eGFR was reduced at 80.5 [IQR: 58–90] mL/min/1.73 m^2^, and median UPCR was increased at 0.3 [IQR: 0.2, 0.5] g/mmol. In terms of induction therapy, 52 (88%) patients received either mycophenolate mofetil (MMF) or cyclophosphamide (CYC), 2 (3%) patients received biologic agent therapy alone, 4 (7%) patients received a combination of biologic agent therapy and immunosuppressants, and 1 (2%) patient received a combination of MMF and Tacrolimus (Table 1).

### 2.2. CRR Attainment at 6, 12, and 24 Months

Data assessing CRR status at all three time points were available for 51 patients (Figure 1). CRR status (at either 6, 12, or 24 months) was attained by 31 patients (61%) at least once. In total, 20 patients (39%) never attained CRR, while 12 patients (24%) were in CRR at all three time points. Overall, 23 (40%), 26 (47%), and 24 (44%) patients were in CRR at 6, 12, and 24 months, respectively (Table 1).

A comparison of patients’ baseline characteristics by CRR status is shown in Table 2. The only significant difference between the CRR 6-month responder and non-responder groups was a lower baseline UPCR in the CRR 6-month responder group. Sixteen (76%) patients in the CRR 6-month responder group were in CRR at 24 months, compared to eight (25%) patients in CRR 6-month non-responder group (Figure 2). Eighteen (69%) patients in the CRR 12-month responder group achieved CRR at 24 months, compared to five (19%) patients in the CRR 12-month non-responder group (Figure 2). Daily glucocorticoid (GC) dose and SLE Disease Activity Index 2000 (SLEDAI-2K) score at 24 months were significantly lower amongst CRR 6-month responders, compared to CRR 6-month non-responders (median GC dose 3.2 mg/day vs. 5.0 mg/day; *p* = 0.014, median SLEDAI-2K 4.0 vs. 6.0; *p* = 0.006). SLEDAI-2K score at 24 months was significantly lower in the CRR 12-month responder group compared with the CRR 12-month non-responder group (4.0 vs. 7.0; *p* = 0.014). The percentage of females was higher in the 12- and 24-month responder groups compared to the non-responder group. Although not statistically significant, the age at diagnosis tended to be higher in the 12- and 24-month responder groups compared to the non-responder group.

### 2.3. Factors Associated with CRR Attainment at 24 Months

In the univariable logistic analysis, achieving CRR at 6 months was significantly associated with the likelihood of CRR achievement at 24 months (odds ratio (OR) = 9.60 (95% confidence interval (CI): 2.66, 34.67), *p* = 0.001; Table 3). In multivariable regression analysis, adjusted for age and antimalarial drug use, CRR at 6 months remained significantly associated with higher CRR achievement at 24 months (OR = 11.23 (95% CI: 2.53, 49.88), *p* = 0.001; Table 3). Similar results were observed in the CRR 12-month responder group (univariable model: OR = 9.45 (95% CI: 2.62, 34.07), *p* = 0.001, multivariable model: OR = 11.39 (95% CI: 2.41, 53.80), *p* = 0.002).

We also evaluated the association between CRR status at 6 months and CRR achievement at 12 months. In univariable logistic analysis, CRR achievement at 6 months was significantly associated with a greater likelihood of CRR achievement at 12 months (OR = 6.93 (95% CI: 2.06, 23.26), *p* = 0.002). In multivariable analysis, CRR 6-month response remained significantly associated with increased CRR achievement at 12 months (OR = 9.72 (95% CI: 2.19, 43.03), *p* = 0.003; Table 3). Details of the univariable and multivariable logistic analyses are shown in Appendix A.

We additionally assessed whether anti-double-stranded DNA (dsDNA) antibody level and C3/C4 levels were associated with CRR attainment. As shown in the Appendix A, no significant associations were observed between subsequent CRR and either anti-dsDNA antibody level and/or C3/C4 levels in our cohort. These findings suggest that, in this adult LN cohort, early renal remission, rather than serological markers, was a stronger predictor of subsequent CRR.

Furthermore, we examined whether early CRR was associated with preservation of kidney function using eGFR slope and UPCR. Specifically, we examined the association between CRR attainment at 6 months and subsequent changes in eGFR and UPCR at 12 and 24 months, as well as between CRR attainment at 12 months and changes in these measures between 12 and 24 months. The results are summarised in Appendix A. We observed that patients achieving CRR at 6 months had improved eGFR between 6 and 24 months, and a slight but statistically significant increase in UPCR between 6 and 12 months. In contrast, CRR attainment at 12 months was not associated with significant changes in either eGFR or UPCR compared with non-responders. Overall, these findings highlight trends that warrant further investigation in larger cohorts to provide more definitive evidence.

## 3. Discussion

In this retrospective observational study, we have demonstrated that achieving CRR within 6 and 12 months of biopsy-proven proliferative LN diagnosis is associated with a significantly greater likelihood of CRR at 24 months, providing evidence that early renal remission is an independent predictor for subsequent CRR attainment among LN patients. There was no association between anti-dsDNA antibody or C3/C4 levels and CRR attainment at 12 or 24 months.

Remission in LN refers to achieving a complete or partial remission of kidney disease activity, defined by clinical criteria [11]. Alongside achieving renal remission, demonstrated by reduced proteinuria and stabilised kidney function [12,13,14], evidence suggests that timing is important, with early remission, typically within 6–12 months of commencing induction therapy, being associated with better clinical outcomes, including higher eGFR, lower rates of renal flares, and prevention of lupus related damage accrual [13,14,15,16]. Our study demonstrated that the odds ratio for achieving CRR at 24 months was similar between patients with CRR status at 6 months and those with CRR at 12 months, consistent with previous studies. In an observational study from Korbet et al., a ≥50% reduction in proteinuria at 6 months predicted long-term renal survival in LN [17]. Inversely, a randomised controlled trial comparing tacrolimus or MMF induction therapy in LN patients found that the absence of renal response at 6 months was independently associated with poor renal survival. An observational study reported that the lack of renal response at 12 months was an independent predictor of poor renal prognosis [4]. Our findings highlight the importance of early evaluation of CRR status at 6–12 months to guide treatment modifications that can impact long-term disease trajectory. In line with this, the EULAR/ERA-EDTA 2019 guidelines recommend reassessment and potential adjustment of therapy if patients do not show at least a partial response by 3 months [18]. Recently published 2024 American College of Rheumatology (ACR) guidelines for the screening, treatment, and management of LN also recommend assessment of renal response within 6 to 12 months of initiating therapy [9]. We identified that patients who achieved CRR status at 6 months had significantly lower baseline proteinuria levels than those who did not attain CRR at 6 months; median UPCR was 0.18 g/mmol compared to 0.42 g/mmol. This is a comparable finding to that by Luis et al., who showed that proteinuria levels of <2 g/day at baseline and at 3 months were predictive of CRR attainment at 12 months, highlighting the importance of diagnosing and treating LN before severe kidney injury has occurred [19]. A recent study reported C3, C4, and anti-dsDNA antibodies as biomarkers for clinical and immunological remission in LN [20]. However, we did not find these complements or antibodies to be associated with CRR attainment. Further validation studies in larger cohorts are warranted.

Maintenance of CRR is an important treatment target that has been associated with lower risks of mortality, CKD, and ESKD. A longitudinal study by Pakchotanon et al. reported that maintaining complete remission for at least five years (defined in their study as proteinuria < 500 mg/day or UPCR < 0.5 g/g and inactive urine sediment) was associated with a lower risk of mortality, ESKD, eGFR <50 mL/min, and doubling of serum creatinine [21]. Gatto et al. showed that complete remission for at least 3 years was required to prevent CKD and overall damage accrual in patients with LN [10]. In contrast, the absence of remission was identified as a predictor of CKD and ESKD [16]. Our findings support the use of early CRR status to predict subsequent CRR status at 24 months, which is in line with the literature describing CRR as a treatment endpoint associated with better long-term kidney outcomes and overall survival.

Our study has several limitations. Given the nature of the retrospective design, we could not accurately evaluate the association between early remission and relapse-free terms. Although our study revealed that the daily GC dose at 24 months was lower in the 6-month CRR responder group compared to the non-responder group, we could not assess time-adjusted GC dose and GC-related organ damage. Further studies are needed to determine whether early remission allows for more rapid GC tapering, leading to reduced GC-related organ damage. This study was conducted at a single centre with a relatively small cohort, and only a few patients received biologic agents. We did not assess histopathological features such as renal amyloidosis or systemic inflammatory markers (e.g., CRP), which may also provide prognostic information, and future studies incorporating these parameters could further refine risk stratification in LN. Future prospective investigations with larger sample sizes are needed to validate the findings of our study. Clinical definitions of complete renal remission may not accurately reflect histological remission of proliferative LN, and we did not have data on repeat kidney biopsies in this cohort to assess this [22]. Furthermore, paediatric LN patients were not included in our study, and evidence to inform LN treatment and outcomes in this population is still lacking [23]. Regardless of these limitations, our study strengthens the utility of using CRR at 6–12 months as an outcome measure not only for hard outcomes such as mortality and ESKD, but also for assessing CRR achievement throughout the clinical course.

## 4. Methods

### 4.1. Study Design and Participants

We conducted a single-centre, retrospective observational study of data collected from patients visiting the Monash Lupus Clinic through the Australian Lupus Registry and Biobank (ALRB), from January 2007 to February 2020. The inclusion criteria were (1) patients who met either the revised American College of Rheumatology (ACR) classification criteria for SLE [24] or the 2012 Systemic Lupus International Collaborating Clinics (SLICC) classification criteria [25] and (2) patients with newly diagnosed, biopsy-proven class III, IV, or III or IV + V LN, according to the International Society of Nephrology/Renal Pathology Society (ISN/RPS) 2003 lupus nephritis classification [26]. Patients who required kidney replacement therapy at the time of kidney biopsy, paediatric (age < 18 years) patients, and patients who had previously been diagnosed with LN were excluded. All patients provided informed consent for their de-identified clinical data to be stored and used for research purposes. This project has been approved by the Monash Health Human Research Ethics Committee (HREC) (project reference number 14262A).

### 4.2. Variables and Definitions

We collected demographic information (including age, sex, ethnicity, smoking history, education level, and SLE duration prior to kidney biopsy and LN diagnosis), histopathological findings of kidney biopsies, and medications used for induction therapy at enrolment. Laboratory data (including serum creatinine, eGFR, UPCR, serum complement levels, and anti-dsDNA level), disease activity scores (measured using the SLE Disease Activity Index—2000; SLEDAI-2K [27]), and daily glucocorticoid (GC) dose were collected at each routine visit. The baseline visit of this study was defined as the date of kidney biopsy performance (or the closest routine visit date to kidney biopsy).

CRR was defined as a reduction in proteinuria to <0.5 g/g (50 mg/mmol) and stabilisation or improvement in eGFR (±10% of baseline) at 6, 12 and 24 months, based on the definition from the Kidney Disease Improving Global Outcomes (KDIGO) 2024 practice guideline for the management of LN [9]. The primary outcome of this study was the achievement of CRR at 24 months.

### 4.3. Statistical Analysis

We described patient characteristics using summary statistics. Continuous variables were reported as median and IQR. Wilcoxon rank-sum test was performed to compare continuous variables, due to the skewed distribution. Categorical variables are shown as numbers and percentages (%) and compared with Pearson’s χ^2^ test. Univariable and multivariable logistic regression analyses were conducted to identify factors associated with CRR. Variables significant in univariable analyses were considered to be potential confounders in multivariable logistic regression models. The results from the analyses were reported as ORs with corresponding 95% CIs. Values of *p* < 0.05 were defined as statistically significant. All analyses were performed using Stata statistcal software (version 18.0).

## 5. Conclusions

Achieving CRR at either 6 or 12 months was associated with an increased likelihood of CRR status at 24 months, with similar magnitudes of association. This study provides evidence that achieving early CRR, within 12 months of commencing induction therapy for class III or IV ± V LN, is a strong independent predictor for attaining subsequent LN remission.

## Figures and Tables

**Figure 1 ijms-26-09634-f001:**
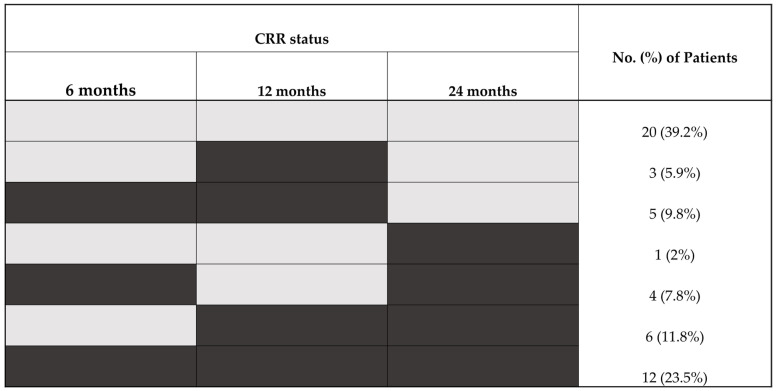
Patient attainment of complete renal response (CRR) at 6, 12, and 24 months (*n* = 51). CRR-responder status is shown in black, and CRR non-responder status in grey.

**Figure 2 ijms-26-09634-f002:**
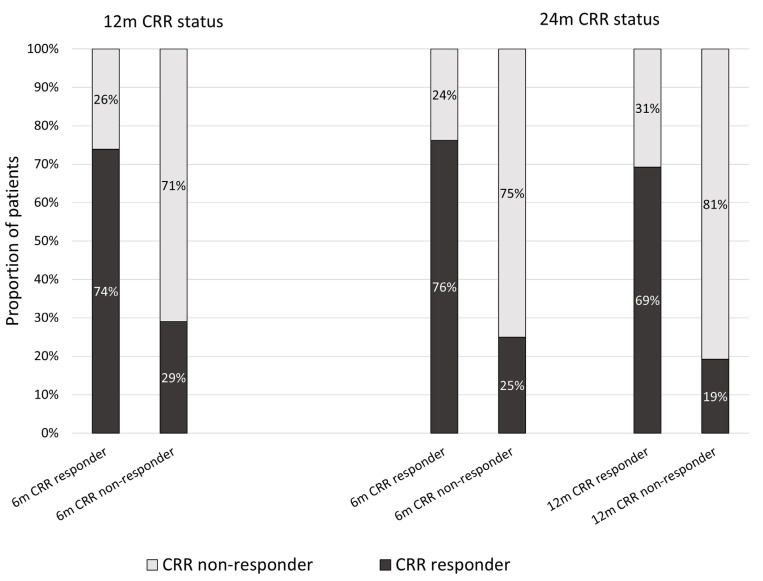
Percentage of subsequent CRR attainment stratified by 6-month and 12-month CRR status.

**Table 1 ijms-26-09634-t001:** Patient characteristics.

	Study Cohort
	Total Number (*n*) of Patients = 60
	*n* (%) or Median [IQR]
**Demographics**	
Female	49 (82%)
Age at LN diagnosis	27 [19, 39]
SLE disease duration prior to biopsy-proven Class III or IV *±* V LN (months)	4.5 [0, 10.5]
Ethnicity	
Caucasian	21 (35%)
Asian	31 (52%)
Others	8 (13%)
Current or ex-smoker ^1^	11 (19%)
Hypertension	12 (20%)
**Baseline Serological profile**	
Anti-dsDNA positive ^2^	53 (93%)
Hypocomplementaemia (low C3/C4) ^3^	47 (80%)
**Kidney Function at LN diagnosis**	
Estimated glomerular filtration rate (eGFR, mL/min/1.73 m^2^) ^4^	80.5 [58, 90]
Urinary protein/creatinine ratio (UPCR, g/mmol)	0.3 [0.2, 0.5]
**Medications**	
Glucocorticoids (GC) exposure at least once (ever)	60 (100%)
Daily GC dose (mg) at:	
Baseline ^5^	17.5 [7.5, 38.8]
6 months ^6^	10.0 [7.5, 15.0]
12 months ^7^	5.0 [1.2, 10.0]
24 months ^8^	5.0 [1.0, 7.5]
Induction therapy ^3^	59 (100%)
MMF	41 (69%)
CYC	11 (19%)
RTX	2 (3%)
MMF/RTX	3 (5%)
MMF/TAC	1 (2%)
CYC/RTX	1 (2%)
Antimalarial therapy ^1^	49 (84%)
**Class of LN (ISN/RPS Classification)**	
III	12 (20%)
IV	26 (43%)
III or IV ± V	22 (37%)
**Complete renal remission (CRR)**	
CRR at least once (CRR-ever) ^9^	31 (61%)
CRR never ^9^	20 (39%)
CRR at 6 months ^1^	23 (40%)
CRR at 12 months ^10^	26 (47%)
CRR at 24 months ^10^	24 (44%)
**SLE Disease Activity Index (SLEDAI)-2K score**	
Baseline ^11^	14.5 [8.0, 19.0]
6 months ^12^	6.0 [4.0, 10.0]
12 months ^13^	6.0 [4.0, 8.0]
24 months ^14^	4.0 [4.0, 8.0]

Data missing for ^1^ 2, ^2^ 3, ^3^ 1, ^4^ 4, ^5^ 20, ^6^ 14, ^7^ 12, ^8^ 8, ^9^ 9, ^10^ 5, ^11^ 22, ^12^ 19, ^13^ 15, ^14^ 11 patients. eGFR = estimated glomerular filtrate rate, CYC, cyclophosphamide; MMF, mycophenolate mofetil, RTX, rituximab; TAC, tacrolimus, UPCR = urine protein to creatinine ratio.

**Table 2 ijms-26-09634-t002:** Comparison of patient characteristics stratified by CRR status at 6, 12 and 24 months.

	CRR at 6 Months			CRR at 12 Months			CRR at 24 Months		
	Non-Responder	Responder		Non-Responder	Responder		Non-Responder	Responder	
	*n* = 35	*n* = 23		*n* = 29	*n* = 26		*n* = 31	*n* = 24	
	*n* (%) or Median [IQR]		*p*-Value	*n* (%) or Median [IQR]		*p*-Value	*n* (%) or Median [IQR]		*p*-Value
**Demographics**									
Female	27 (77%)	20 (87%)	0.35	19 (66%)	25 (96%)	0.005	23 (74%)	23 (96%)	0.031
Age at LN diagnosis	28 [17, 40]	27 [19, 39]	0.60	23 [15, 31]	30 [24, 39]	0.088	24 [15, 30]	28.5 [21.5, 41.5]	0.064
Pre-biopsy SLE duration (months)	4 [0, 12]	5 [0, 9]	0.90	5 [1, 12]	4.5 [0.0, 9.0]	0.30	7 [1, 13]	3 [0, 7.5]	0.083
Asian ethnicity	21 (60%)	11 (48%)	0.49	16 (55%)	13 (50%)	0.70	16 (52%)	15 (62%)	0.85
Current or ex-smoker ^1^	7 (21%)	4 (17%)	0.72	7 (25%)	3 (12%)	0.20	6 (20%)	3 (12%)	0.46
**Serological profile**									
Anti-dsDNA positive ^2^	32(97%)	20(87%)	0.15	26 (96%)	23 (88%)	0.28	26 (93%)	22 (92%)	0.87
Hypocomplementaemia ^3^	26(76.5%)	19(82.6%)	0.57	22 (79%)	20 (77%)	0.84	23 (77%)	19 (79%)	0.83
LN histological Class (ISN/RPS 2003 classification)			0.64			0.17			0.82
III	6 (17%)	6 (26%)		5 (17%)	7 (27%)		7 (23%)	4 (17%)	
IV	15 (43%)	10 (43%)		9 (31%)	12 (46%)		12 (39%)	11 (46%)	
III/IV + V	14 (40%)	7 (30%)		15 (52%)	7 (27%)		12 (39%)	9 (38%)	
eGFR (mL/min/1.73 m^2^) ^4^	90 [59, 90]	77 [60, 90]	0.45	84 [57, 90]	82 [60, 90]	0.98	90 [57, 90]	77 [60, 90]	0.92
UPCR (g/mmol)	0.42 [0.27, 0.61]	0.18 [0.12, 0.30]	0.003	0.30 [0.14, 0.59]	0.27 [0.15, 0.48]	0.55	0.41 [0.20, 0.61]	0.23 [0.13, 0.43]	0.058
**Medications**									
Glucocorticoids (GC) ever	35 (100%)	23 (100%)		29 (100%)	26 (100%)		31 (100%)	24 (100%)	
Daily GC dose (mg) at:									
Baseline ^5^	15 [7, 40]	20 [8, 38]	0.8	15 [5, 40]	20 [8, 40]	0.57	20 [10, 25]	15 [5, 40]	0.59
6 months ^6^	10 [5, 15]	10 [9, 15]	0.54	10 [5, 15]	10 [8, 15]	0.66	10 [5, 15]	11.2 [9, 15]	0.29
12 months ^7^	5 [0, 10]	5 [1, 10]	0.88	5 [4, 13]	5 [1, 10]	0.27	7.5 [2, 15]	5 [1, 5]	0.11
24 months ^8^	5 [3, 10]	3.2 [0, 5]	0.014	5 [1, 7.5]	2.5 [0, 5]	0.079	5 [2.5, 10.0]	4 [0, 5.]	0.038
Induction therapy ^3^			0.59			0.84			0.70
MMF	23 (68%)	16 (70%)		20 (69%)	18 (69%)		21 (68%)	18 (75%)	
CYC	6 (18%)	5 (22%)		6 (21%)	5 (19%)		7 (23%)	3 (12%)	
RTX	2 (6%)	0 (0%)		1 (3.4%)	1 (3.8%)		1 (3%)	1 (4%)	
MMF/RTX	2 (6%)	1 (4%)		1 (3.4%)	1 (3.8%)		1 (3%)	1 (4%)	
MMF/Tac	1 (3%)	0 (0%)		1 (3.4%)	0 (0%)		1 (3%)	0 (0%)	
CYC/RTX	0 (0%)	1 (4%)		0 (0%)	1 (3.8%)		0 (0%)	1 (4%)	
Antimalarial therapy ^1^	22 (67%)	12 (52%)	0.27	24 (86%)	23 (88%)	0.76	20 (65%)	13 (54%)	0.44
**Complete renal remission (CRR)**									
CRR at 6 months ^1^				6 (21%)	17 (65%)	0.001	5 (17%)	16 (67%)	<0.001
CRR at 12 months ^9^	9 (29%)	17 (74%)	0.001				8 (28%)	18 (78%)	<0.001
CRR at 24 months ^9^	8 (25%)	16 (76%)	<0.001	5 (19%)	18 (69%)	<0.001			
**SLEDAI-2K score**									
Baseline ^10^	10 [8, 16]	14.5 [10, 19]	0.41	12.5 [8, 16]	13 [8, 18]	0.95	10 [8, 16]	13 [8, 20]	0.37
6 months ^11^	8 [6, 14]	4 [2, 4]	<0.001	8 [5, 10]	4 [2, 6]	0.008	7 [4, 12]	4 [2, 8]	0.040
12 months ^12^	8 [6, 10]	4 [2, 4]	0.011	8 [6.5, 10]	4 [2, 4]	<0.001	8 [6, 10]	4 [2, 4]	0.002
24 months ^13^	6 [4, 10]	4 [4, 4]	0.005	7 [4, 12]	4 [4, 4]	0.012	8 [6, 10]	4 [3, 4]	<0.001

Data missing for ^1^ 2, ^2^ 3, ^3^ 1, ^4^ 4, ^5^ 20, ^6^ 14, ^7^ 12, ^8^ 8, ^9^ 5, ^10^ 22, ^11^ 19, ^12^ 15, ^13^ 11 patients. eGFR = estimated glomerular filtrate rate, CYC, cyclophosphamide; MMF, mycophenolate mofetil, RTX, rituximab; SLEDAI-2K = SLE Disease Activity Index-2000, Tac, tacrolimus, UPCR = urine protein: creatinine ratio.

**Table 3 ijms-26-09634-t003:** Association of CRR responder status at 6 and 12 months with subsequent CRR attainment.

	12 m CRR Achievement		24 m CRR Achievement	
	Univariable Logistic Regression Analysis	Multivariable Logistic Regression Analysis	Univariable Logistic Regression Analysis	Multivariable Logistic Regression Analysis
	OR (95%CI), *p*-Value	OR ^1^ (95%CI), *p*-Value	OR (95%CI), *p*-Value	OR ^1^ (95%CI), *p*-Value
6 months non-responder	1.00	1.00	1.00	1.00
6 months responder	6.93 (2.06, 23.26), 0.002	9.72 (2.19, 43.03), 0.003	9.60 (2.66, 34.67), 0.001	11.23(2.53, 49.88), 0.001
12 months non-responder			1.00	1.00
12 months responder			9.45 (2.62, 34.07), 0.001	11.39 (2.41, 53.80), 0.002

^1^ ORs adjusted for age (category < 20, 20–40, >40) and antimalarial drugs use.

## Data Availability

The datasets used in this study are not publicly available due to the strict governance policies of the Australian Lupus Registry & Biobank (ALRB). Access to deidentified data may be considered upon reasonable request, and subject to ALRB’s data access policy.

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
