# Peer review of "Early Renal Remission Is Associated with Increased Likelihood of Subsequent Remission in Lupus Nephritis: Single-Centre Observational Study in Australia"

_ijms, 2025, doi:10.3390/ijms26199634_

Round 1
Reviewer 1 Report
Comments and Suggestions for Authors
See the attached file

Author Response
Can you add biopsy figures to the study?
Response: The study cohort, comprising 60 patients with newly diagnosed, kidney biopsy-confirmed class III or IV±V LN, was selected from 369 patients treated at Monash Health Lupus Clinic between 2013 and 2020 and enrolled in the Australian Lupus Registry and Biobank (ALRB). To clarify this point, we have added the following sentence to the Results section:
(line 52-55)
“Of the 369 patients with SLE treated at the Monash Health Lupus Clinic, a tertiary-level hospital servicing the South-East of Melbourne, Australia, and enrolled in the Australian Lupus Registry and Biobank (ALRB) between 2013 and 2020, 60 (16%) patients had a new diagnosis of kidney biopsy-confirmed class III or IV ± V LN and comprised the study cohort.”
How can we correlate the following article, which discussed renal pathology, with the prognosis of SLE in children? Pennesi M, Benvenuto S. Lupus Nephritis in Children: Novel Perspectives. Medicina (Kaunas).2023 Oct 16;59(10):1841. doi: 10.3390/medicina59101841. PMID: 37893559; PMCID: PMC10607957.
Response: We thank the reviewer for drawing attention to the article by Pennesi and Benvenuto (2023), which provides a comprehensive overview of renal pathology and emerging perspectives in paediatric lupus nephritis (LN). Our study was conducted in an adult patient cohort, and therefore paediatric LN was outside the scope of our analysis. We agree that paediatric LN remains under-studied and that treatment and outcome data in this population are limited. To acknowledge this important gap, we have included the following statement in the Discussion section under limitations, with reference to the above article:
(line 172-174)
“Furthermore, paediatric LN patients were not included in our study, and evidence to inform LN treatment and outcomes in this population is still lacking.”
What is the advantage of your paper over this paper that examined anti-double-stranded DNA as a prognostic marker? De Vriese AS, Sethi S, Fervenza FC. Lupus nephritis: redefining the treatment goals. Kidney Int. 2025 Feb;107(2):198-211. doi: 10.1016/j.kint.2024.10.018. Epub 2024 Nov 8. PMID: 39521057.
Response: Once again, we thank the Reviewer for citing this paper by De Vriese et al, which highlights three concurrent treatment goals in LN: (i) clinical remission assessed using markers of renal inflammation, including haematuria, proteinuria, eGFR, and complements, (ii) immunologic remission using dsDNA as a biomarker, and (iii) preservation of kidney function using eGFR slope. After considering the Reviewer’s comment, we further assessed the association of CRR with anti-dsDNA antibody level and C3/C4 levels. As shown in the table below (Table S3), we did not find associations between subsequent CRR and dsDNA levels, or C3/C4 levels, in our cohort. On this basis, we concluded this paper focusing on the early renal remission, defined based on stable eGFR and improvement in proteinuria, as a predictor for subsequent CRR. Thus, the novelty of our study lies in demonstrating that early renal remission, rather than serological markers, predicts subsequent outcomes in our adult cohort. We have amended the Results and the Discussion sections to reflect this.
Added section in the Results:
(line 94-98)
“We additionally assessed whether anti-double-stranded DNA (dsDNA) antibody level and C3/C4 levels were associated with CRR attainment. As shown in the Supplementary Materials (Table S3), no significant associations were observed between subsequent CRR and either anti-dsDNA antibody level or C3/C4 levels in our cohort. These findings suggest that, in this adult LN cohort, early renal remission, rather than serological markers, was a stronger predictor of subsequent CRR.”
Added sentence in the Discussion:
(line 129-130)
“There was no association between anti-dsDNA antibody or C3/C4 levels and CRR attainment at 12 or 24 months.”
Table S3: Association of CRR achievement at 24 months with anti-double-stranded DNA (dsDNA) antibody level and C3/C4 levels at 6 and 12 months in univariable logistic analysis
|
|
12m CRR achievement |
24m CRR achievement |
|
|
OR (95%CI), p-value |
OR (95%CI), p-value |
|
dsDNA level at baseline (+1 IU/mL) |
1.00 (1.00, 1.00), 0.15 |
1.00 (1.00, 1.00), 0.36 |
|
dsDNA level at 6 months (+1 IU/mL) |
1.01 (1.00, 1.00), 0.47 |
1.00 (0.99, 1.00), 0.31 |
|
dsDNA level at 12 months (+1 IU/mL) |
|
1.00 (1.00, 1.00), 0.69 |
|
dsDNA negative at 6 months |
1.00 |
1.00 |
|
dsDNA positive at 6 months |
0.61 (0.09, 4.01), 0.61 |
2.87 (0.28, 29.68), 0.38 |
|
|
|
|
|
dsDNA negative at 12 months |
|
1.00 |
|
dsDNA positive at 12 months |
|
1.32 (0.20, 8.64), 0.77 |
|
|
|
|
|
C3 level at baseline |
0.55 (0.08, 3.86), 0.55 |
0.33 (0.04, 2.45), 0.28 |
|
C3 level at 6 months |
2.00 (0.18, 22.26), 0.57 |
2.47 (0.19, 32.43), 0.49 |
|
C3 level at 12 months |
|
2.28 (0.18, 28.21), 0.52 |
|
|
|
|
|
C4 level at baseline |
0.01 (0.00, 8.24), 0.17 |
0.01 (0.00, 11.84), 0.20 |
|
C4 level at 6 months |
2.13 (0.03, 140.11), 0.72 |
0.11 (0.00, 27.31), 0.43 |
|
C4 level at 12 months |
|
0.66 (0.00, 1568.73), 0.92 |
|
|
6months complete responders RC* (95% CI), p-value |
12months complete responders RC* (95% CI), p-value |
|
eGFR difference between: |
|
|
|
6 & 12 months |
6.69 (-0.45, 13.84), 0.080 |
|
|
6 & 24 months |
8.29 (0.48, 16.09), 0.038 |
|
|
12 & 24 months |
-0.20 (-4.79, 4.39), 0.93 |
0.96 (-3.32, 5.25), 0.65 |
|
|
|
|
|
UPCR difference between: |
|
|
|
6 & 12 months |
0.09 (0.01, 0.17), 0.026 |
|
|
6 & 24 months |
0.04 (-0.08, 0.15), 0.52 |
|
|
12 & 24 months |
-0.05 (-0.18, 0.08), 0.45 |
-0.01 (-0.14, 0.13), 0.91 |
*RC = regression coefficient between responders and non-responders adjusted for age & anti-malarial therapy
You mentioned ‘However, reliable clinical markers for predicting sustained renal remission remain poorly understood in patients 47 with LN. This study aims to determine whether early renal remission (ERR) is associated with subsequent sustained 48 renal remission’. Can we consider GFR, Renal function, and antibody assay in combination as reliable markers of remission in comparison with proteinuria”
Response: We assume that the Reviewer was suggesting to consider dsDNA antibody levels in combination with eGFR and UPCR to determine remission. We did not include dsDNA antibody levels in the definition of complete renal remission as dsDNA is considered as a less reliable marker of renal function in LN due to poor correlation with renal pathology, high inter-patient variability and high fluctuating and non-specific nature of the assays. As shown in Table 2, the proportions of dsDNA+ve patients in both responder and non-responder groups remained high (>90%) at all three time points (6-, 12- & 24 months) without statistical significance.
In line with the Reviewer’s suggestion, we, however, performed additional analysis to evaluate whether dsDNA positivity or C3/C4 levels were associated with renal remission at 12months and 24 months (supplementary tables 2 & 3). These analyses did not demonstrate any significant association between these serological markers and CRR attainment in our cohort.
We have acknowledged this in the Results and the Discussion sections in following sentences:
(line 94-98)
‘We additionally assessed whether anti-double-stranded DNA (dsDNA) antibody level and C3/C4 levels were associated with CRR attainment. As shown in the Supplementary Materials (Table S3), no significant associations were observed between subsequent CRR and either anti-dsDNA antibody level or C3/C4 levels in our cohort. These findings suggest that, in this adult LN cohort, early renal remission, rather than serological markers, were stronger predictor of subsequent CRR.’
(line 151-153)
'A recent study has reported C3, C4 and anti-dsDNA antibodies as biomarkers for clinical and immunological remission in LN [20]. However, we did not find these complements or anti-dsDNA antibodies to be associated with CRR attainment. Further validation studies in larger cohorts are warranted.’
Do your cases have hypertension? It is not clear in the inclusion criteria
Response: Yes, approximately 12% of the study cohort had hypertension. Hypertension was not included as part of the inclusion criteria. We have now added the number and proportion of patients with hypertension in Table 1 for clarity.
You checked by histopathology the process of amyloidosis in the kidney, a marker of bad prognosis. Have checked the inflammatory markers in the blood, for example, CRP, to predict the prognosis
Response: We did not assess histopathological evidence of amyloidosis on kidney biopsy specimens as inflammatory markers such as CRP were not available for analysis. We acknowledge the importance of identifying additional prognostic biomarkers and have highlighted this as a limitation of our study.
(line 167-169)
‘We did not assess histopathological features such as renal amyloidosis or systemic inflammatory markers (e.g., CRP), which may also provide prognostic information, and future studies incorporating these parameters could further refine risk stratification in LN’

Reviewer 2 Report
Comments and Suggestions for Authors
There is increasing evidence that early treatment sufficient to achieve remission improves long term outcomes in rheumatic disease. The authors have retrospectively reviewed the renal outcomes in 60 patients with lupus nephritis in a single centre to ascertain predictors of outcome at 24 months. The cohort was reflective of standard lupus cohorts ie predominantly female (82%), young (median age 27 years), mostly dsDNA+ve (93%), with low complements in the majority (80%) and all had biopsy proven disease. Disease duration was short (median 4.5 months) and the majority were on steroids.
The only factor appearing to predict response at 6 months was baseline urinary PCR – lower PCR predicting response. Being in renal remission at 6 months predicted remission at 12 and 24 months; similarly being in remission at 12 month predicted remission at 24 months.
The study is singe centre and numbers are relatively small. Only very small numbers received rituximab or cyclophosphamide and there appears to be no differences between the groups in terms of therapies.
Minor – the colours used for the charts and bar charts (fig 1 and 2) do not project well when printed in B&W – you may want to review this pre-publication.
This is presumably your journal style – but we usually expect to see Abstract, followed by introduction, then methods, results, discussion and conclusion. I thought it was a bit odd to have methods at the end!
Author Response
There is increasing evidence that early treatment sufficient to achieve remission improves long term outcomes in rheumatic disease. The authors have retrospectively reviewed the renal outcomes in 60 patients with lupus nephritis in a single centre to ascertain predictors of outcome at 24 months. The cohort was reflective of standard lupus cohorts ie predominantly female (82%), young (median age 27 years), mostly dsDNA+ve (93%), with low complements in the majority (80%) and all had biopsy proven disease. Disease duration was short (median 4.5 months) and the majority were on steroids.
The only factor appearing to predict response at 6 months was baseline urinary PCR – lower PCR predicting response. Being in renal remission at 6 months predicted remission at 12 and 24 months; similarly, being in remission at 12 months predicted remission at 24 months.
The study is single centre and numbers are relatively small. Only very small numbers received rituximab or cyclophosphamide and there appears to be no differences between the groups in terms of therapies.
Response: We acknowledge the small sample size and single-centre design as limitations. We have added the following sentence to the Discussion under limitation to emphasize this point.
‘This study was conducted at a single-centre with a relatively small cohort, and few patients received biologic agents.’
The colours used for the charts and bar charts (fig 1 and 2) do not project well when printed in B&W – you may want to review this pre-publication.
Response: We have revised the Figures 1 & 2 to black-and-white format to ensure clarity and readability when printed.
Figure 1: Patient attainment of complete renal response (CRR) at 6-, 12- and 24-months (n=51). CRR-responder status is shown in Black, and CRR non-responder status in Grey
Figure 2: Percentage of subsequent CRR attainment stratified by 6 months- and 12 months CRR status
This is presumably your journal style – but we usually expect to see Abstract, followed by introduction, then methods, results, discussion and conclusion. I thought it was a bit odd to have methods at the end!
Response – Yes, the manuscript has been formatted according to the Journal’s style, which places the Methods section after the Discussion.
